# The Relationship Between Economic Pressure, Emotional Neglect and Anxiety Among Chinese Left-Behind Children: Based on Response Surface Analysis

**DOI:** 10.3390/bs15121679

**Published:** 2025-12-04

**Authors:** Suxia Liao, Danyun Wang, Kuo Zhang, Jingxin Wang

**Affiliations:** 1Department of Social Psychology, School of Sociology, Nankai University, Tianjin 300350, China; 1120241280@mail.nankai.edu.cn; 2Academy of Psychology and Behavior, Faculty of Psychology, Tianjin Normal University, Tianjin 300387, China; wangjingxin@tjnu.edu.cn

**Keywords:** economic pressure, emotional neglect, perceived discrimination, anxiety, response surface analysis

## Abstract

Based on the cumulative risk model and cognitive appraisal theory, this study examined the complex relationships between economic pressure, emotional neglect, and anxiety among left-behind children (LBC), focusing on the mediating role of perceived discrimination and nonlinear risk patterns. A cross-sectional survey was conducted with 618 LBC (aged 11–16 years) using standardized scales. Polynomial regression combined with Response Surface Analysis (RSA) was utilized to analyze congruence and incongruence effects. The results revealed that under congruent conditions, the association of economic pressure and emotional neglect with anxiety showed a marginally significant nonlinear accelerating trend, whereas their prediction of perceived discrimination followed a linear trend. Under incongruent conditions, emotional neglect demonstrated a stronger independent predictive effect on anxiety compared to economic pressure. Furthermore, perceived discrimination partially mediated the relationships between these risk factors and anxiety. These findings validate the cumulative risk model within the LBC context, demonstrating that risk factors operate in complex, non-additive ways. This highlights the necessity for differentiated interventions and suggests that reshaping LBC’s subjective cognitive appraisals is key to reducing anxiety.

## 1. Introduction

Left-behind children defined as minors under the age of 16 living in rural China who cannot live with their parents due to one or both parents working away from home ([51]), represent a substantial and vulnerable subpopulation. According to educational statistics released by the Ministry of Education in August 2023, the number of left-behind children in compulsory education reached 15.5056 million. Their unique upbringing predisposes them to mental health challenges ([9]), significantly increasing the likelihood of emotional, behavioral, and social problems ([16]). Existing research on left-behind children’s mental health has predominantly focused on depression as a core indicator, with limited attention to anxiety symptoms. This may be because depressive symptoms are more salient and typical, while anxiety, as a common emotional experience, is often dismissed as “normal growing pains” or perceived as a comorbid symptom of depression ([4]), leading to its neglect by guardians and teachers. However, recent surveys reveal an anxiety detection rate of 33.24% among LBC, significantly exceeding that of the general child population ([17]; [37]). Beyond its immediate impairment of cognitive and social functioning ([2]; [17]). Childhood anxiety exerts detrimental effects across multiple developmental domains. Cognitively, it impedes functioning, leading to academic deficits and behavioral maladjustment ([2]); socially, it erodes interpersonal skills, increasing susceptibility to isolation and peer victimization ([17]). Crucially, childhood anxiety is a robust predictor of adult psychopathology, predisposing individuals to severe outcomes such as 2, suicidal ideation, and substance abuse ([11]). Therefore, identifying the drivers and mechanisms of anxiety in LBC is crucial for establishing effective early screening and intervention strategies.

The psychological adversities faced by left-behind children in China are inextricably linked to environmental stressors. Notably, family economic hardship and emotional neglect constitute primary ecological risks and are robust predictors of depressive symptoms ([53]; [59]). However, research examining the relationships between these two risk factors and anxiety in left-behind children (LBC) remains relatively scarce, and the underlying mechanisms warrant further exploration. Driven by structural imbalances in urban-rural economic development, LBC generally experience low household income. This not only subjects them to an objective deprivation of material resources but also imposes significant psychological stress. Family economic pressure is defined as the subjective perception of stress that arises when a family’s resources fail to meet its needs and demands ([56]). Empirical studies indicate that children exposed to greater economic pressure are more susceptible to negative emotions, conduct disorders, substance abuse, and academic failure ([6]; [24]; [41]). Furthermore, the Family Stress Model posits that family economic pressure indirectly impacts children’s psychological adjustment through disrupted emotional and behavioral processes ([42]). Consequently, family economic pressure may serve as a critical precipitating factor for anxiety symptoms in LBC.

In addition, left-behind children are susceptible to parental emotional neglect associated with prolonged parental absence. Emotional neglect refers to parents’ failure to meet children’s fundamental needs for support, belonging, and emotional care, characterized by insensitivity to their distress and disregard for their social and emotional development ([54]). Emotional neglect is characterized by its covert and persistent nature, rendering its negative impact on mental health more severe than overt abuse ([64]). According to attachment theory, while children may receive care from other guardians, emotional neglect from primary attachment figures (i.e., parents) impedes the formation of secure attachment, fostering insecure internal working models—negative self-perceptions and expectations of others as untrustworthy ([8]). This suggests that emotional neglect may constitute a significant risk factor for anxiety in left-behind children ([23]; [27]).

Most existing studies have independently examined the relationship between economic pressure or emotional neglect and the mental health of left-behind children ([34]; [62]), limiting the ability to explain their synergistic effects on psychological well-being. Given the unique developmental context of left-behind children, economic pressure and emotional neglect represent two major risk factors affecting their mental health at the material and emotional levels, respectively, embodying the risks of family resource deprivation and family relationship disruption they face ([60]). However, their mechanisms of influence on mental health may differ significantly: emotional neglect primarily operates through psychological pathways such as attachment insecurity and impaired self-worth ([7]; [35]), while economic pressure more directly threatens their basic survival needs, educational opportunities, and future expectations, thereby inducing existential anxiety ([42]). According to the Cumulative Risk Model, children exposed to dual or multiple concurrent risks exhibit the poorest developmental outcomes ([47]; [57]). Crucially, this model posits that such risks interact synergistically, rather than merely additively to exacerbate maladjustment. Yet, previous studies have frequently relied on traditional linear regression frameworks, thereby oversimplifying these complex dynamics. Such conventional methods fail to model nonlinear effects (e.g., whether risk accumulation accelerates anxiety) or assess outcomes when risk levels are incongruent (i.e., unbalanced). Consequently, the present study adopts Response Surface Analysis (RSA), a methodological approach uniquely suited to rigorously disentangle the linear and nonlinear mechanisms linking these dual risks to anxiety.

Furthermore, the mechanisms linking economic pressure and emotional neglect to anxiety in left-behind children (LBC) warrant further exploration. While prior studies have identified mediators such as trait mindfulness or shame ([30]; [61]), these investigations have largely overlooked the LBC population, thereby neglecting the crucial role of their subjective appraisal of adversity. Cognitive Appraisal Theory posits that an individual’s subjective evaluation of stressors determines the psychological impact of stressful events ([33]). In this context, perceived discrimination, defined as the subjective perception of unfair treatment directed at oneself or one’s group offers an integrative explanatory framework ([44]). Specifically, LBC are uniquely vulnerable to societal stigmatization, which heightens their sensitivity to identity threats ([58]). Empirical evidence indicates that LBC report significantly higher levels of perceived discrimination compared to their non-left-behind peers ([49]). Consequently, when confronted with objective resource deprivation stemming from economic pressure (e.g., consumption limitations), LBC are prone to attributing these deficits to societal exclusion rather than neutral environmental factors ([19]). Concurrently, prolonged emotional neglect fosters maladaptive cognitive schemas characterized by hypervigilance and an anticipation of social rejection ([13]). This schema predisposes children to interpret ambiguous social cues through a hostile, self-referential lens ([21]). Synthesizing these pathways, a left-behind child with a negative self-concept derived from emotional neglect is more likely to perceive the resource scarcity caused by economic pressure as a specific manifestation of discriminatory exclusion ([18]). Therefore, we hypothesize that economic pressure and emotional neglect exacerbate anxiety in LBC by increasing their perceived discrimination.

Current research has yet to integrate these ecological risks and cognitive mechanisms into a unified model using nonlinear analytics. Therefore, the primary objective of this study is to examine the joint predictive effects of economic pressure and emotional neglect on anxiety among left-behind children using RSA, and to investigate the mediating role of perceived discrimination ([55]; [26]). By clarifying these complex relationships, this study aims to expand the explanatory framework of the Cumulative Risk Model and provide empirical evidence for targeted, differentiated interventions. Therefore, we propose the following hypotheses:Hypothesis 1: Under congruence, anxiety among left-behind children increases progressively with rising economic pressure and emotional neglect.Hypothesis 2: Under incongruence, the effects of “high economic pressure—low emotional neglect” and “low economic pressure—high emotional neglect” on left-behind children’s anxiety differ.Hypothesis 3: Perceived discrimination will mediate the relationships between economic pressure, emotional neglect, and anxiety among left-behind children.

The proposed conceptual model is illustrated in Figure 1.

## 2. Method

### 2.1. Participants

Participants were recruited via cluster sampling from four public secondary schools (two junior high and two senior high schools) in Hebei Province, China. The recruitment followed a distinct two-stage process. First, to minimize selection bias, questionnaires were distributed to all students in randomly selected classes within these schools. A total of 4000 questionnaires were distributed, and 3940 responses were returned. Secondly, from this pool, we identified students meeting the inclusion criteria for left-behind children (LBC): (1) being under 16 years of age; and (2) having one or both parents who had migrated for work for a minimum of six months. Initially, 709 students met these criteria. After excluding participants with incomplete responses or those who failed attention checks, a final analytic sample of 618 left-behind children was obtained (effective response rate within the target group = 87.2%). The final sample consisted of 321 boys (51.9%) and 297 girls (48.1%). Participants ranged in age from 11 to 16 years, with an average age of 14.67 years old. Regarding parental migration status, 350 participants (56.6%) had only their fathers working away, 16 (2.6%) had only their mothers working away, and 252 (40.8%) had both parents working away. In terms of separation duration, 288 children had been separated for less than 2 years, 169 for 2 to 5 years, 69 for 6 to 10 years, and 82 for 10 years or more.

### 2.2. Ethics Procedures

This study was approved by the Ethics Committee of Nankai University (Approval No. NKUIRB2024096). All procedures performed were in accordance with the 1964 Helsinki Declaration and its later amendments.

### 2.3. Procedure

Given the involvement of minors, a rigorous informed consent process was implemented. Prior to data collection, written informed consent was obtained from the parents or legal guardians of all participants. Additionally, verbal assent was obtained from the adolescents themselves immediately before the survey commenced. Participants were explicitly informed that their participation was voluntary and anonymous, and that they retained the right to withdraw at any time without penalty.

To ensure the validity of responses, particularly regarding sensitive topics such as emotional neglect and mental health, adolescents completed the questionnaires independently in the classroom in the absence of their parents. This setting served to minimize social desirability bias and potential parental influence on self-disclosure. For the formal assessment, paper-and-pencil questionnaires were administered during a designated class period. To maintain standardization across different grades and schools, a uniform protocol was adopted. Specifically, a trained psychology graduate student was present in each classroom to read standardized instructions and clarify any items as needed. This administration method, consistent with the students’ routine testing procedures, ensured procedural uniformity for all participants regardless of age or grade level.

### 2.4. Measurement Questionnaire

For all measurement tools employed in this study (including the Economic Pressure Scale, Emotional Neglect Scale, Perceived Discrimination Scale, and GAD-7), the mean score of the items was calculated to represent the level of each variable, with higher scores indicating higher levels of the respective construct.

#### 2.4.1. Economic Pressure

Economic pressure was measured using the Economic Pressure Scale developed by [56] ([56]). This scale consists of four items that assess financial hardship related to basic necessities, such as clothing, food, housing, and transportation. For example, “My family doesn’t have enough money to buy new clothes,” and “My family doesn’t have enough money to buy the food I like.” Responses were rated on a 5-point scale, ranging from 1 (*Never*) to 5 (*Always*). The average of the four items was calculated, with higher scores indicating greater family economic hardship. The Cronbach’s α coefficient for the Economic Pressure Scale in this study was 0.82.

#### 2.4.2. Emotional Neglect

Emotional neglect was measured using the 10-item Neglect subscale of the Child Psychological Abuse and Neglect Scale ([12]). The subscale, which includes one validity check item, assesses experiences of neglect with items such as, “My parents don’t care about the changes in my academic performance.” Responses are rated on a 5-point scale ranging from 0 (*Never*) to 4 (*Always*), with higher scores indicating greater severity of emotional neglect. The Cronbach’s α coefficient is 0.88 for the total scale and 0.83 for the Neglect subscale, as reported in the original validation study. The Cronbach’s α of the Emotional Neglect Scale in this study was 0.73.

#### 2.4.3. Perception of Discrimination

The Perceived Discrimination Questionnaire, developed by [49] ([49]), was used to assess children’s perceived discrimination. This scale was adapted from the Perceived Outgroup Rejection Scale ([46]) and the Perceived Discrimination Questionnaire ([31]). The 6-item questionnaire is rated on a 5-point scale and comprises two subscales: personal-level perceived discrimination and group-level perceived discrimination. Higher total scores indicate a greater perception of discrimination. The Cronbach’s α coefficients for the personal-level and group-level subscales were 0.81 and 0.82, respectively. The Cronbach’s α coefficient for the Perceived Discrimination Questionnaire in this study was 0.87.

#### 2.4.4. Anxiety

Children’s anxiety symptoms were assessed using the 7-item Generalized Anxiety Disorder scale (GAD-7; [50]). Items are rated on a 4-point scale ranging from 0 (*Not at all*) to 3 (*Nearly every day*). A total score was calculated by summing the responses to all items, resulting in a score range of 0~21. Higher scores indicate greater anxiety severity. The Cronbach’s α coefficient for the scale in the present study was 0.91. The Cronbach’s α of the GAD-7 in this study was 0.89.

### 2.5. Statistical Methods

Descriptive statistics, correlation analyses, and *t*-tests were performed using JASP 0.19. Polynomial regression and response surface analysis (RSA) were conducted using the *RSA* package in R (version 4.2.2). Following the procedure outlined by [26] ([26]), we first grand-mean centered the predictor variables, economic pressure and emotional neglect, to mitigate multicollinearity. Subsequently, anxiety and perceived discrimination were each regressed on the five polynomial terms (i.e., the linear, quadratic, and interaction terms of the predictors) to yield the following regression equation:*Z* = *b*_0_ + *b*_1_*X* + *b*_2_*Y* + *b*_3_*X*^2^ + *b*_4_*X**Y* + *b*_5_*Y*^2^ + *e*
where *X* represents economic pressure and *Y* represents emotional neglect. The terms *X*^2^, *Y*^2^, and *XY* are the respective squared terms and the interaction term.

Furthermore, we employed the block variable approach described by [14] ([14]) to examine the mediating role of perceived discrimination in the relationship between economic pressure, emotional neglect, and symptoms of anxiety. Specifically, a weighted linear composite (i.e., the block variable) was created by combining the five polynomial regression terms (*X*, *Y*, *X*^2^, *XY*, and *Y*^2^). The weights for these terms were their corresponding coefficients from the initial regression analysis. The polynomial regression was then re-run to obtain the path coefficient from the block variable to the mediator (perceived discrimination). The indirect effect was calculated as the product of this path and the path from the mediator to the outcome (anxiety). To test the significance of this mediation, we generated 95% confidence intervals for the indirect effect using 5000 bootstrap samples.

## 3. Result

### 3.1. Common Method Bias Test

As all data were collected from student self-reports, we conducted a Harman’s single-factor test to assess for potential common method bias. An exploratory factor analysis was performed on all items from the four questionnaires. The results revealed seven factors with eigenvalues greater than 1, and the first unrotated factor accounted for 13.80% of the total variance. As this is below the critical threshold of 40%, these results suggest that common method bias was not a serious concern in this study.

### 3.2. The Correlation Between Economic Stress, Emotional Neglect, Perception of Discrimination and Anxiety

As shown in Table 1, emotional neglect, economic pressure, perceived discrimination, and anxiety were all significantly and positively intercorrelated (all *p*s < 0.001). Additionally, significant gender differences were found in levels of anxiety, and age was significantly correlated with all four variables. Therefore, gender and age were included as control variables in all subsequent analyses. In addition, the present study employed the 7-item Generalized Anxiety Disorder scale (GAD-7) to screen for anxiety symptoms among Chinese left-behind children. To prioritize screening sensitivity and maximize the identification of potentially high-risk individuals who may require professional evaluation, a cutoff score of 8 was selected. This threshold aligns with the core public health principle of minimizing missed diagnoses ([29]; [32]). Although a cutoff of 10 is more commonly utilized in clinical and research settings ([50]), the 8-point criterion better serves the screening objectives of this study. Based on this standard, the detection rate of anxiety symptoms among left-behind children in the study sample was 40.29%.

### 3.3. Polynomial Regression and Response Surface Analysis

To determine the suitability of the data for polynomial regression and response surface analysis (RSA), we first analyzed the distribution of sample responses. The results showed that 37.70% of the sample (*N* = 233) was congruent on emotional neglect and economic pressure, while 28.97% (*N* = 179) had higher emotional neglect than economic pressure, and 33.33% (*N* = 206) had lower emotional neglect than economic pressure. As each category exceeded the 10% minimum threshold, the data were deemed suitable for the intended analyses.

The parameters for the polynomial regression and RSA models predicting anxiety and Perception of discrimination are presented in Table 2. Additionally, multicollinearity diagnostics were performed for the models predicting perceived discrimination and anxiety. All Variance Inflation Factor (VIF) values were well below 3, ranging from 1.21 to 2.27, indicating that multicollinearity was not a concern.

### 3.4. Response Surface Analysis of Economic Pressure, Emotional Neglect, and Perceived Discrimination

First, we examined whether perceived discrimination among left-behind children was higher when economic pressure and emotional neglect were congruently high versus congruently low. The results revealed a significant, positive slope for the line of congruence (Y = X), a_1_ = 0.65, *p* < 0.001, 95%CI = [0.55, 0.75], indicating that perceived discrimination increased as the two predictors rose in agreement. The curvature along this line was not significant, a_2_ = 0.03, *p* = 0.588, 95%CI = [−0.07, 0.12], suggesting this relationship is linear. For the line of incongruence (Y = −X), the slope was not significant, a_3_ = 0.002, *p* = 0.981, 95%CI = [−0.14, 0.15], which indicates no significant difference in perceived discrimination between the conditions of high economic pressure/low emotional neglect and low economic pressure/high emotional neglect (Figure 2). The curvature of this line was not significant, a_4_ = 0.07, *p* = 0.453, 95%CI = [−0.11, 0.25], indicating no significant nonlinear relationship along the line of incongruence.

### 3.5. Response Surface Analysis of Economic Pressure, Emotional Neglect, and Anxiety

We then examined the surface related to anxiety. Along the line of congruence, the analysis yielded a significant positive slope (a_1_ = 0.48, *p* < 0.001, 95%CI = [0.37, 0.60]), demonstrating that anxiety levels were highest when both economic pressure and emotional neglect were high. The analysis of the curvature (a_2_ = 0.10, *p* = 0.071, 95%CI = [−0.01, 0.21]) indicated a marginally significant nonlinear trend along this line. Furthermore, we analyzed the line of incongruence. The slope was significant and negative (a_3_ = −0.20, *p* = 0.024, 95%CI = [−0.38, −0.03]), implying that children reported higher anxiety in the condition of low economic pressure and high emotional neglect than in the opposing condition (Figure 3). The curvature of this line was not significant (a_4_ = 0.03, *p* = 0.759, 95%CI = [−0.16, 0.23]), indicating no significant nonlinear relationship along the line of incongruence.

### 3.6. The Mediating Effect of Perceived Discrimination

Finally, we investigated the mediating role of perceived discrimination in the relationship between the combined effects of economic pressure, emotional neglect, and anxiety. As shown in Figure 4, Gender and age were included as covariates in the two regression equations of the mediation model. When predicting the perceived discrimination, age showed a significant positive effect (*B* = 0.09, *se* = 0.03, *p* = 0.008, 95%CI = [0.02, 0.15]), while the effect of gender was not significant (*B* = 0.003, *se* = 0.03, *p* = 0.934, 95%CI = [−0.06, 0.07]). In the prediction of the anxiety, gender emerged as a significant positive predictor (*B* = 0.12, *se* = 0.03, *p* < 0.001, 95%CI = [0.06, 0.18]). Age showed a marginally significant positive effect (*B* = 0.05, *se* = 0.03, *p* = 0.087, 95%CI = [−0.01, 0.11]). After controlling for the mediator, the direct effect of the combined predictor block variable on anxiety remained significant (*B* = 0.14, *se* = 0.04, *p* = 0.001, 95%CI = [0.05, 0.22]). The effect of the block variable on perceived discrimination (*B* = 0.59, *se* = 0.03, *p* < 0.001, 95%CI = [0.53, 0.65]), as was the effect of perceived discrimination (the mediator) on anxiety was also significant (*B* = 0.56, *se* = 0.03, *p* < 0.001, 95%CI = [0.49, 0.62]). A bootstrapping procedure with 5000 samples was used to test the indirect effect. The results revealed a significant indirect effect of the combined predictors on anxiety through perceived discrimination (Indirect Effect = 0.33, *se* = 0.03, *p* < 0.001, 95%CI = [0.28, 0.38]). This mediating effect accounted for 70% of the total effect.

## 4. Discussion

This study employed polynomial regression and response surface analysis (RSA) to investigate the predictive effects of economic pressure and emotional neglect on anxiety among left-behind children, while examining the mediating role of perceived discrimination. Moving beyond traditional single-risk and simple cumulative-risk research paradigms, the response surface analysis examined the synergistic impact patterns of economic pressure and emotional neglect under congruence and incongruence. Theoretically, this study extends the application of cumulative risk model within left-behind children’s populations, offering a refined framework to explain how dual dilemmas in economic resources and emotional environments translate into profound psychological distress ([5]; [39]). Practically, the findings provide critical insights for developing targeted intervention strategies addressing anxiety among left-behind children. Future interventions should adopt differentiated approaches based on the distinct predictive patterns of economic pressure and emotional neglect experienced by left-behind children. Additionally, the significant mediating role of perceived discrimination suggests that intervention priorities can shift from addressing objective adversities to reshaping their subjective cognitive appraisals. By guiding them to actively reinterpret social environments, interventions may effectively reduce anxiety through cognitive restructuring.

This study reveals that economic pressure and emotional neglect, as two typical ecological risks faced by left-behind children, significantly predict anxiety. Specifically, under congruence, the predictive effects of economic pressure and emotional neglect on anxiety exhibited a marginally significant nonlinear accelerated growth pattern (*p* = 0.071). This tentative finding suggests that as levels of both stressors increase synchronously, left-behind children’s anxiety may worsen at an accelerating rate rather than through simple cumulative effects ([3]). Although this potential trend aligns with the “threshold effect” proposed by the Stress-Vulnerability Model, which posits that when individual vulnerability and environmental stress interact beyond a critical threshold, internalizing problems like anxiety escalate sharply ([28]). Under incongruence, polynomial regression and response surface analysis demonstrate that emotional neglect exerted a stronger predictive effect than economic pressure. Unlike economic pressure, which primarily operates through indirect pathways affecting parental emotions and parenting styles ([20]), emotional neglect directly disrupts parent–child attachment formation, leading to maladaptive self-concepts such as feelings of unworthiness and being unloved ([10]; [45]; [63]). It is important to acknowledge that, consistent with attachment theory, left-behind children often develop secondary attachments to surrogate caregivers (e.g., grandparents) during parental absence. While these secondary attachments can serve as a protective buffer against severe developmental issues ([1]), our findings suggest that for many children, this compensatory caregiving may not fully offset the specific loss of the primary parental bond. Consequently, the persistent emotional deficit and the objective limitations of surrogate caregiving often lead to an elevated subjective perception of emotional neglect. By compromising attachment security and self-worth, emotional neglect exerts a more profound impact on left-behind children’s anxiety than economic pressure. These findings hold significant implications for targeted interventions. First, the potential accelerating trend of anxiety underscores the possible urgency of early detection and comprehensive, multidimensional psychological interventions to prevent symptom escalation. Second, the higher weighting of emotional neglect indicates that intervention strategies should prioritize strengthening parent–child communication and providing stable emotional support to effectively buffer or even reduce anxiety risks among left-behind children.

This study further examined the predictive effects of economic pressure and emotional neglect on perceived discrimination among left-behind children. Results indicated that the effect sizes for economic pressure and emotional neglect in predicting perceived discrimination are consistent. Moreover, under congruence, left-behind children experiencing high economic pressure compounded by high emotional neglect exhibited the highest levels of perceived discrimination, these findings suggest that interventions aiming to mitigate perceived discrimination should simultaneously address both economic pressure and emotional neglect. The prediction of perceived discrimination by economic pressure and emotional neglect followed a linear additive pattern. This finding aligns with the core tenet of the Cumulative Risk Model, which posits that multiple risk factors synergistically produce more severe negative outcomes than single risks ([34]). Notably, the linear additive pattern of the two stressors in predicting left-behind children’s perceived discrimination contrasts sharply with their marginally significant nonlinear accelerating trend in predicting anxiety. This discrepancy suggests fundamental differences in how left-behind children’s perceived discrimination versus anxiety symptoms respond to risk accumulation. Anxiety, as a physiological and emotional response, aligns with the tipping point hypothesis within Allostatic Load Theory ([43]). The simultaneous presence of severe economic deprivation and emotional voids may overwhelm the child’s psychological regulatory system, leading to a systemic collapse where symptoms escalate exponentially once a tolerance threshold is breached ([22]). In contrast, perceived discrimination functions as a cognitive tally of social devaluation. Consistent with Social Identity Theory ([52]), this appraisal likely follows a cumulative cue process, where economic poverty and emotional neglect serve as distinct but additive sources of visible stigma. Each additional risk factor linearly increases the salience of being different or inferior relative to peers, without necessarily requiring a threshold to be crossed to trigger the cognitive recognition of discrimination ([38]). Under incongruence conditions (high economic pressure—low emotional vs. low economic pressure—high emotional neglect), no significant differences in perceived discrimination were detected. Given that differential response patterns to risk factors often imply distinct intervention strategies ([15]), findings indicate that interventions targeting perceived discrimination in left-behind children should differ from those addressing anxiety symptoms.

Perceived discrimination was found to mediate the relationships between economic pressure, emotional neglect, and anxiety among left-behind children. Specifically, the indirect effect of perceived discrimination (b = 0.33) was significantly stronger than the direct effects of economic pressure and emotional neglect (b = 0.14), accounting for 70% of the total effect. This finding underscores the pivotal role of social-cognitive factors in the psychopathology of LBC. Drawing on Resilience Theory, which conceptualizes resilience as a dynamic process of positive adaptation to adversity ([40]), we propose that when LBC interpret family adversities as evidence of social rejection (i.e., high perceived discrimination), their adaptive capacity is compromised, rendering them more susceptible to anxiety. Conversely, maintaining low levels of perceived discrimination despite family hardships functions as a critical protective factor against internalizing disorders ([25]). This suggests that children’s subjective interpretations of their social environment may serve as more proximal predictors of internalizing problems than objective family hardships ([36]; [48]). Practically, these findings highlight a novel and actionable avenue for intervention. Unlike traditional approaches that often focus on intractable family economic conditions or complex parent–child dynamics, our results suggest that school and social environments offer more accessible points of intervention. By implementing anti-discrimination curricula, enhancing teacher sensitivity, and fostering inclusive peer cultures, educators can effectively disrupt the pathway linking perceived discrimination to anxiety. This advocates for a strategic shift: moving from exclusively repairing family environments to restructuring cognitive appraisals, thereby offering a timely and effective approach to improving mental health.

Finally, the present study has several limitations. First, its cross-sectional design, wherein all variables were measured at a single point in time, is effective for revealing associations but precludes the establishment of causal relationships. Future research should employ longitudinal or cross-lagged panel designs to track these variables across multiple time points, which would allow for a more definitive test of the causal pathways proposed in this study. Second, all core variables, emotional neglect, economic pressure, perceived discrimination, and anxiety were measured through child self-report questionnaires. This reliance on a single data source may introduce shared method variance. Future studies could enhance the reliability and validity of the findings by incorporating multi-source data, such as assessments from parents or teachers, or by including more objective indicators like household income. Third, it is important to acknowledge that the nonlinear effect on anxiety was only marginally significant. While RSA requires substantial statistical power to detect curvilinear effects, future studies with larger sample sizes are recommended to further verify the robustness of this accelerating trend.

## 5. Conclusions

By employing response surface analysis, this study elucidates the complex, synergistic effects of economic pressure and emotional neglect on anxiety among left-behind children. The findings reveal that these dual risks operate through a nonlinear, accelerating mechanism to exacerbate anxiety, with emotional neglect proving more detrimental than economic pressure under incongruent conditions. Furthermore, perceived discrimination serves as a key mediator linking these ecological stressors to mental health outcomes. Collectively, this study validates the cumulative risk model in the context of left-behind children and highlights a critical practical implication: effective interventions must transcend material support to prioritize the restoration of emotional connections and the reshaping of cognitive appraisals, thereby mitigating the profound impact of family adversity on anxiety.

## Figures and Tables

**Figure 1 behavsci-15-01679-f001:**
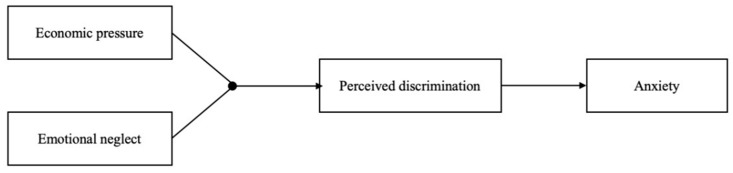
The proposed conceptual model based on the Cumulative Risk Model ([57]) and Cognitive Appraisal Theory ([33]).

**Figure 2 behavsci-15-01679-f002:**
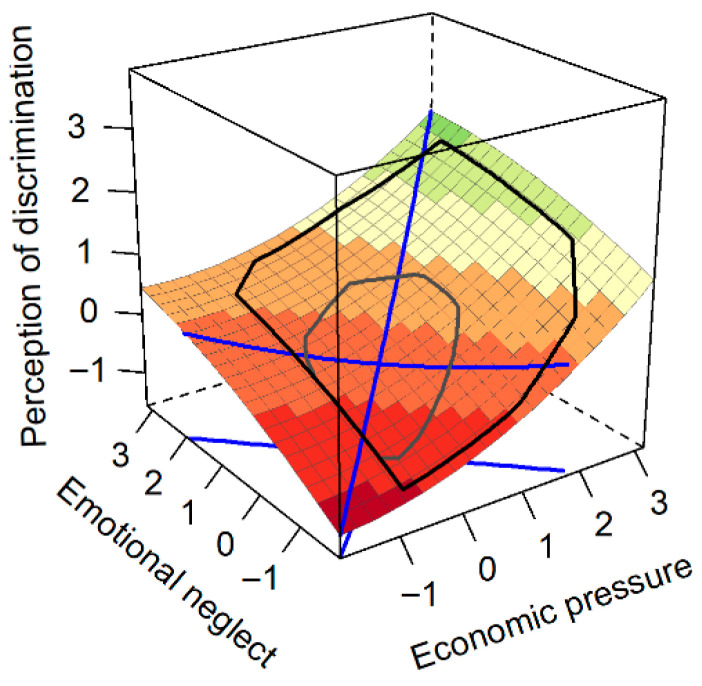
Prediction of Perceived Discrimination from Economic Pressure and Emotional Neglect, the blue lines represent the line of congruence (Y = X) and the line of incongruence (Y = −X).

**Figure 3 behavsci-15-01679-f003:**
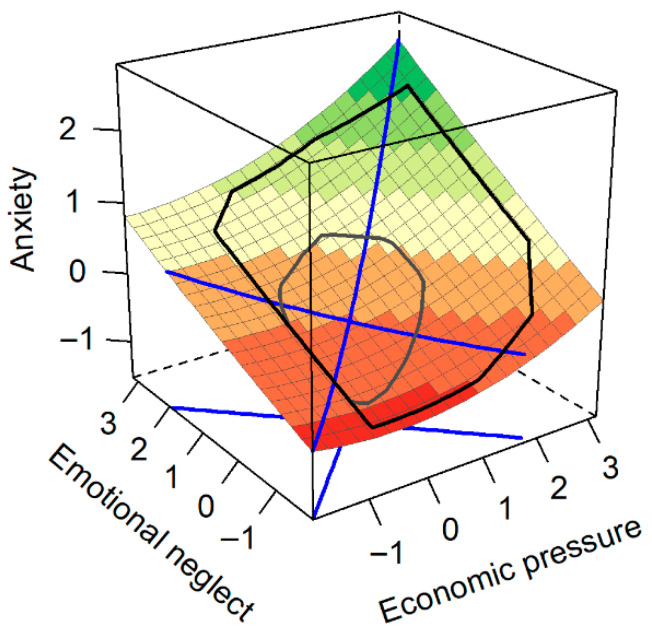
Prediction of Anxiety from Economic Pressure and Emotional Neglect, the blue lines represent the line of congruence (Y = X) and the line of incongruence (Y = −X).

**Figure 4 behavsci-15-01679-f004:**
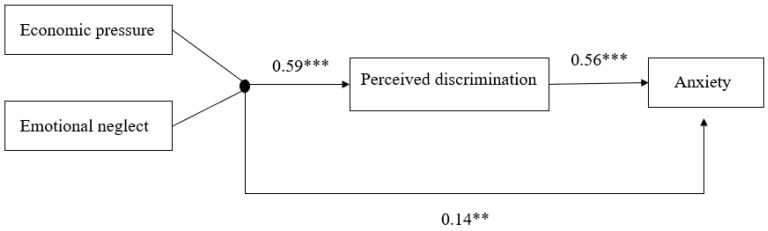
The mediating effect of perceived discrimination in the relationship between the joint cumulative risks (economic pressure and emotional neglect) and anxiety. Note: The model was tested using the block variable approach. ** *p* < 0.01, *** *p* < 0.001.

**Table 1 behavsci-15-01679-t001:** Correlation analysis results of emotional neglect, economic stress, Perception of discrimination and anxiety.

Variable	Gender	Age	Economic Pressure	Emotional Neglect	Perception of Discrimination	Anxiety
Gender	1					
Age	0.001	1				
Economic pressure	−0.05	0.19 ***	1			
Emotional neglect	−0.03	0.11 **	0.33 ***	1		
Perception of discrimination	−0.03	0.19 ***	0.49 ***	0.49 ***	1	
Anxiety	0.10 *	0.18 ***	0.31 ***	0.42 ***	0.64 ***	1
*M* ± *SD*		14.67 ± 1.25	1.72 ± 0.81	1.15 ± 0.65	1.15 ± 0.73	1.05 ± 0.68

Note: * *p* < 0.05, ** *p* < 0.01, *** *p* < 0.001, the same below. Quantification method for adolescent gender: male = 1, female = 2.

**Table 2 behavsci-15-01679-t002:** Polynomial regression and response surface analysis parameters of perceptual discrimination and anxiety caused by economic stress and emotional neglect. Note: *** *p* < 0.001.

Variable	Perception of Discrimination			Anxiety		
	Estimate	*SE*	*p*	Estimate	*SE*	*p*
Intercept (b_0_)	−0.07	0.05	0.153	−0.08	0.05	0.108
gender	0.21	0.07	0.002	−0.001	0.06	0.992
age	0.10	0.03	<0.001	0.08	0.03	0.002
Economic pressure (b_1_)	0.33	0.05	<0.001	0.14	0.06	0.023
Emotional neglect (b_2_)	0.33	0.04	<0.001	0.34	0.04	<0.001
Economic pressure^2^ (b_3_)	0.08	0.04	0.043	0.06	0.05	0.193
Emotional neglect × Economic pressure (b_4_)	−0.02	0.05	0.666	0.04	0.06	0.519
Emotional neglect^2^ (b_5_)	−0.03	0.03	0.327	0.01	0.04	0.877
Slope of the Line of congruence (a_1_)	0.65	0.05	<0.001	0.48	0.06	<0.001
Curvature of the Line of congruence (a_2_)	0.03	0.05	0.588	0.10	0.06	0.071
Slope of the line of incongruence (a_3_)	0.03	0.07	0.981	−0.20	0.09	0.024
Curvature of the Line of incongruence (a_4_)	0.07	0.09	0.453	0.03	0.10	0.759
*R* ^2^	0.23 ***			0.35 ***		

## Data Availability

Some or all data, models, or code generated or used during the study are available from the corresponding author by request.

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
