# Peer review of "The Relationship Between Economic Pressure, Emotional Neglect and Anxiety Among Chinese Left-Behind Children: Based on Response Surface Analysis"

_behavsci, 2025, doi:10.3390/bs15121679_

Round 1

Reviewer 1 Report

Comments and Suggestions for Authors

The study was conducted based on the cumulative risk model and cognitive appraisal theory, expanding the explanatory framework of relevant theories within the group of left-behind children. It offered a new perspective and provided fresh ideas for research in this field. The results relatively clearly revealed the complex relationships between family economic pressure, emotional neglect, perceived discrimination, and anxiety in left-behind children under different conditions. It clarified the mediating mechanism of perceived discrimination, providing empirical evidence for formulating differentiated intervention strategies for left-behind children, and thus held certain practical value.

there are still some areas for improvement in the study:

1.Research Methodology: Participant Description

Problem: The description of the participants lacks sufficient detail and clarity. The manuscript does not specify the educational level of the participants (e.g., junior high school, senior high school, or a mix). Furthermore, key demographic information such as the participants’ average age and gender distribution is not reported, which limits the reader’s ability to assess the sample’s characteristics and the generalizability of the findings.

Suggestion: To enhance transparency, the authors should supplement the Participants section (2.1) with the following information: (a) the specific grade levels or educational stage of the students; (b) the mean age and standard deviation of the sample; and © the gender breakdown (e.g., N and percentage for males and females).

2. Measurement Tools: Scoring Procedures

Problem: The scoring procedures for the measurement tools are not consistently reported. The manuscript clearly states that the “average of the four items was calculated” for the Economic Pressure Scale. However, for the Emotional Neglect, Perceived Discrimination, and Anxiety scales, it is not specified whether the total score or the mean score was used in the analyses. This ambiguity affects the study’s replicability.

Suggestion: To improve methodological rigor, the authors should explicitly state the scoring method for all core variables. This could be achieved by adding a clarifying sentence at the beginning of section 2.3 (e.g., “Scores for the Emotional Neglect and Perceived Discrimination scales were calculated as the mean of all items, while the total score was used for the GAD-7 anxiety scale.”).

3. Section 2.3.2: Reliability of the Emotional Neglect Scale

Problem: The reporting of reliability for the Emotional Neglect scale is incomplete. The text cites the Cronbach’s α from the “original validation study” but fails to report the reliability coefficient for this scale within the current sample. Since reliability is sample-dependent, it is crucial to report it for the data actually used in the study.

Suggestion: The authors must add the Cronbach’s α coefficient for the Emotional Neglect subscale as calculated in the present study. As a best practice, I also recommend verifying and reporting the reliability coefficients for all scales based on the current sample to ensure comprehensive methodological reporting.

4. Response Surface Analysis Results: Interpretation of Marginal Significance

Problem: The interpretation of the non-linear RSA result for anxiety warrants more caution. The curvature along the line of congruence for anxiety is marginally significant (a2 = 0.10, p = 0.071). The manuscript presents this as a definitive “non-linear accelerating trend” in the Abstract and Discussion, which may overstate the certainty of the finding.

Suggestion: In the Discussion section (e.g., 4.1), the authors should rephrase this finding using more cautious language, such as “a marginally significant non-linear accelerating trend (p = 0.071).” It would be appropriate to frame this as an exploratory finding that warrants confirmation in future research with larger samples.

5. Discussion

Problem: The discussion regarding the differing predictive patterns for anxiety versus perceived discrimination lacks theoretical depth. The authors correctly identify that anxiety follows a non-linear pattern while perceived discrimination follows a linear one, suggesting “fundamental differences” in their response mechanisms. However, these potential mechanisms are not elaborated upon, which weakens this part of the discussion and misses an opportunity for a significant theoretical contribution.

Suggestion: To strengthen this section, the authors should elaborate on the theoretical reasons for these distinct patterns.

6. other suggestion

Statistical Reporting: Include 95% CIs for all key coefficients (e.g., mediation effects).

Language: Minor grammatical errors (e.g., “emotion neglect” → “emotional neglect”; “anxious” → “anxiety symptoms”).

In the Statistical Methods section (2.4), the authors state they employed the block variable approach to “examine the mediating role of relative deprivation.” However, throughout the rest of the manuscript, including the model figure (Fig. 4), the results section (3.6), and the discussion, the mediator is consistently referred to as perceived discrimination.

Conclusion: This study presents valuable findings on a vulnerable population. The points raised above are intended to help the authors refine their manuscript and maximize its contribution to the field. I believe that with these revisions, the paper will be a strong candidate for publication.

Comments on the Quality of English Language

Language: Minor grammatical errors (e.g., “emotion neglect” → “emotional neglect”; “anxious” → “anxiety symptoms”).

Author Response

The study was conducted based on the cumulative risk model and cognitive appraisal theory, expanding the explanatory framework of relevant theories within the group of left-behind children. It offered a new perspective and provided fresh ideas for research in this field. The results relatively clearly revealed the complex relationships between family economic pressure, emotional neglect, perceived discrimination, and anxiety in left-behind children under different conditions. It clarified the mediating mechanism of perceived discrimination, providing empirical evidence for formulating differentiated intervention strategies for left-behind children, and thus held certain practical value.

Response:

We are sincerely grateful to the reviewer for your thorough review and valuable feedback. The comments provided were extremely helpful in identifying areas for improvement and refining our arguments. We have made every effort to address the concerns raised and believe that the manuscript has been substantially strengthened as a result. Our detailed responses to each comment are listed below. To facilitate your review, our responses are presented in blue text, and the specific revisions made to the manuscript are marked in red text.

there are still some areas for improvement in the study:

Comment 1:

Research Methodology: Participant Description

Problem: The description of the participants lacks sufficient detail and clarity. The manuscript does not specify the educational level of the participants (e.g., junior high school, senior high school, or a mix). Furthermore, key demographic information such as the participants’ average age and gender distribution is not reported, which limits the reader’s ability to assess the sample’s characteristics and the generalizability of the findings.

Suggestion: To enhance transparency, the authors should supplement the Participants section (2.1) with the following information: (a) the specific grade levels or educational stage of the students; (b) the mean age and standard deviation of the sample; and © the gender breakdown (e.g., N and percentage for males and females).

Response:

We sincerely thank the reviewer for pointing out this oversight. We agree that detailed demographic information is essential for assessing the sample’s characteristics and the generalizability of the findings. In accordance with your suggestion, we have revised the "Participants" (Section 2.1) section to explicitly state the educational stages (junior and senior high school), the gender breakdown, and the age.

“Participants were recruited via cluster sampling from four public secondary schools (two junior high and two senior high schools) in Hebei Province, China. The recruitment followed a distinct two-stage process. First, to minimize selection bias, questionnaires were distributed to all students in randomly selected classes within these schools. A total of 4,000 questionnaires were distributed, and 3,940 responses were returned. Secondly, from this pool, we identified students meeting the inclusion criteria for left-behind children (LBC): (1) being under 16 years of age; and (2) having one or both parents who had migrated for work for a minimum of six months. Initially, 709 students met these criteria. After excluding participants with incomplete responses or those who failed attention checks, a final analytic sample of 618 left-behind children was obtained (effective response rate within the target group = 87.2%). The final sample consisted of 321 boys (51.9%) and 297 girls (48.1%). Participants ranged in age from 11 to 16 years, with an average age of 14.67 years old. Regarding parental migration status, 350 participants (56.6%) had only their fathers working away, 16 (2.6%) had only their mothers working away, and 252 (40.8%) had both parents working away. In terms of separation duration, 288 children had been separated for less than 2 years, 169 for 2 to 5 years, 69 for 6 to 10 years, and 82 for 10 years or more. ”(page4, line 171 to 187)

Comment 2:

Measurement Tools: Scoring Procedures

Problem: The scoring procedures for the measurement tools are not consistently reported. The manuscript clearly states that the “average of the four items was calculated” for the Economic Pressure Scale. However, for the Emotional Neglect, Perceived Discrimination, and Anxiety scales, it is not specified whether the total score or the mean score was used in the analyses. This ambiguity affects the study’s replicability.

Suggestion: To improve methodological rigor, the authors should explicitly state the scoring method for all core variables. This could be achieved by adding a clarifying sentence at the beginning of section 2.3 (e.g., “Scores for the Emotional Neglect and Perceived Discrimination scales were calculated as the mean of all items, while the total score was used for the GAD-7 anxiety scale.”).

Response:

We appreciate this helpful suggestion to improve methodological rigor. We have revised Section 2.3 to explicitly state the scoring method for all variables. Specifically, we clarified that the mean score was calculated for all measures to ensure consistency and comparability.

“For all measurement tools employed in this study (including the Family Economic Pressure Scale, Emotional Neglect Scale, Perceived Discrimination Scale, and GAD-7), the mean score of the items was calculated to represent the level of each variable, with higher scores indicating higher levels of the respective construct.” (page 5, line 210~213)

Comment 3:

Section 2.3.2: Reliability of the Emotional Neglect Scale

Problem: The reporting of reliability for the Emotional Neglect scale is incomplete. The text cites the Cronbach’s α from the “original validation study” but fails to report the reliability coefficient for this scale within the current sample. Since reliability is sample-dependent, it is crucial to report it for the data actually used in the study.

Suggestion: The authors must add the Cronbach’s α coefficient for the Emotional Neglect subscale as calculated in the present study. As a best practice, I also recommend verifying and reporting the reliability coefficients for all scales based on the current sample to ensure comprehensive methodological reporting.

Response:

We thank the reviewer for this crucial reminder. We have revised Section 2.3.2 to include the Cronbach’s α coefficient for all scales are based on the present study's data.

“The Cronbach's α coefficient for the Economic Pressure Scale in this study was 0.82.” (page 5, line 211~222)

“The Cronbach's α of the Emotional Neglect Scale in this study was 0.73.” (page 6, line 230~231)

“The Cronbach's α coefficient for the Perceived Discrimination Questionnaire in this study was 0.87.” (page 6, line 240~241)

“The Cronbach's α of the GAD-7 in this study was 0.89.” (page 6, line 247~248)

Comment 4:

Response Surface Analysis Results: Interpretation of Marginal Significance

Problem: The interpretation of the non-linear RSA result for anxiety warrants more caution. The curvature along the line of congruence for anxiety is marginally significant (a2 = 0.10, p = 0.071). The manuscript presents this as a definitive “non-linear accelerating trend” in the Abstract and Discussion, which may overstate the certainty of the finding.

Suggestion: In the Discussion section (e.g., 4.1), the authors should rephrase this finding using more cautious language, such as “a marginally significant non-linear accelerating trend (p = 0.071).” It would be appropriate to frame this as an exploratory finding that warrants confirmation in future research with larger samples.

Response:

We sincerely thank the reviewer for the valuable insights regarding statistical rigor. We agree that the p = 0.071 indicates marginal significance, and we acknowledge that our previous interpretation may have overstated the certainty of this finding. Accordingly, we have revised the Abstract and Discussion sections of the manuscript to reflect a more cautious interpretation.

Abstract

“Under congruent conditions, the association of economic pressure and emotional neglect on anxiety showed a marginally significant nonlinear accelerating trend” (page 1, line 21)

Discussion

“Specifically, under congruence, the predictive effects of economic pressure and emotional neglect on anxiety exhibited a marginally significant non-linear accelerated growth pattern (p = 0.071). This tentative finding suggests that as levels of both stressors increase synchronously, left-behind children’s anxiety may worsen at an accelerating rate rather than through simple cumulative effects (Appleyard et al., 2005). Although this potential trend aligns with the "threshold effect" proposed by the Stress-Vulnerability Model” (page 10, line 392~398)

“These findings hold significant implications for targeted interventions. First, the potential accelerating trend of anxiety underscores the possible urgency of early detection and comprehensive” (page 10, line 416~417)

“Third, it is important to acknowledge that the non-linear effect on anxiety was only mar-ginally significant. While RSA requires substantial statistical power to detect curvilinear effects, future studies with larger sample sizes are recommended to further verify the robustness of this accelerating trend.” (page 12, line 487~490)

Comment 5:

Discussion

Problem: The discussion regarding the differing predictive patterns for anxiety versus perceived discrimination lacks theoretical depth. The authors correctly identify that anxiety follows a non-linear pattern while perceived discrimination follows a linear one, suggesting “fundamental differences” in their response mechanisms. However, these potential mechanisms are not elaborated upon, which weakens this part of the discussion and misses an opportunity for a significant theoretical contribution.

Suggestion: To strengthen this section, the authors should elaborate on the theoretical reasons for these distinct patterns.

Response:

We appreciate this constructive comment. We agree that a deeper theoretical explanation regarding the differing predictive patterns is necessary. In response, we have revised the Discussion section to elaborate on the distinct mechanisms driving anxiety (non-linear) versus perceived discrimination (linear), drawing upon relevant psychological theories to support these interpretations.

“Anxiety, as a physiological and emotional response, aligns with the tipping point hypothesis within Allostatic Load Theory (McEwen, 1998). The simultaneous presence of severe economic deprivation and emotional voids may overwhelm the child’s psychological regulatory system, leading to a systemic collapse where symptoms escalate exponentially once a tolerance threshold is breached (Guidi et al., 2020). In contrast, perceived discrimination functions as a cognitive tally of social devaluation. Consistent with Social Identity Theory (Tajfel & Turner, 1979), this appraisal likely follows a cumulative cue process, where economic poverty and emotional neglect serve as distinct but additive sources of visible stigma. Each additional risk factor linearly increases the salience of being different or inferior relative to peers, without necessarily requiring a threshold to be crossed to trigger the cognitive recognition of discrimination (Liu et al., 2023).”(page 11, line 438~449)

Tajfel, H., & Turner, J. C. (2001). An integrative theory of intergroup conflict. In M. A. Hogg & D. Abrams (Eds.), Intergroup relations: Essential readings (pp. 94–109). Psychology Press.

Liu, Xie, T., Li, W., Tao, Y., Liang, P., Zhao, Q., & Wang, J. (2023). The relationship between perceived discrimination and wellbeing in impoverished college students: a moderated mediation model of self-esteem and belief in a just world. Current Psychology, 42(8), 6711–6721. https://doi.org/10.1007/s12144-021-01981-4

McEwen, B. S. (1998). Protective and damaging effects of stress mediators. New England journal of medicine, 338(3), 171–179. https://doi.org/10.1056/NEJM199801153380307

Guidi, J., Lucente, M., Sonino, N., & Fava, G. A. (2020). Allostatic load and its impact on health: a systematic review. Psychotherapy and Psychosomatics, 90(1), 11–27. https://doi.org/10.1159/000510696

Comment 6:

other suggestion

Statistical Reporting: Include 95% CIs for all key coefficients (e.g., mediation effects).

Language: Minor grammatical errors (e.g., “emotion neglect” → “emotional neglect”; “anxious” → “anxiety symptoms”).

In the Statistical Methods section (2.4), the authors state they employed the block variable approach to “examine the mediating role of relative deprivation.” However, throughout the rest of the manuscript, including the model figure (Fig. 4), the results section (3.6), and the discussion, the mediator is consistently referred to as perceived discrimination.

Conclusion: This study presents valuable findings on a vulnerable population. The points raised above are intended to help the authors refine their manuscript and maximize its contribution to the field. I believe that with these revisions, the paper will be a strong candidate for publication.

Response:

We represent our sincere gratitude for the reviewer’s detailed observations and encouraging conclusion. We have addressed these points as follows: 

Statistical Reporting: We have added 95% confidence intervals (CIs) for all key coefficients, including the mediation effects, to the revised results.

“Finally, we investigated the mediating role of perceived discrimination in the relationship between the combined effects of economic pressure, emotional neglect, and anxi-ety. As shown in Fig. 4, after controlling for the mediator, the direct effect of the combined predictor block variable on anxiety remained significant (B = 0.14, se = 0.09, p = 0.001, 95%CI = [0.10, 0.19]). The effect of the block variable on perceived discrimination (B = 0.60, se = 0.07, p < 0.001, 95%CI = [0.55, 0.65]), as was the effect of perceived discrimination (the mediator) on anxiety was also significant (B = 0.57, se = 0.04, p < 0.001, 95%CI = [0.53, 0.63]). A bootstrapping procedure with 5,000 samples was used to test the indirect effect. The re-sults revealed a significant indirect effect of the combined predictors on anxiety through perceived discrimination (Indirect Effect = 0.33, p < 0.001, 95%CI = [0.30, 0.38]). This mediating effect accounted for 70% of the total effect.” (page 9, line 359~365).

  1. Language: We have corrected the specified grammatical errors (e.g., "emotional neglect", "anxiety symptoms") and thoroughly proofread the manuscript.

“Through response surface analysis (RSA), we examined the relationships between family economic pressure, emotional neglect, and anxiety in left-behind children…”(page 1, line 12)

“with limited attention to anxiety symptoms” (page 2, line 45)

  1. Consistency: We apologize for the clerical error in Section 2.4. We have corrected the term "relative deprivation" to "perceived discrimination" to ensure consistency with the rest of the manuscript and the conceptual model.

“Subsequently, anxiety and perceived discrimination were each regressed on the five polynomial terms (i.e., the linear, quadratic, and interaction terms of the predictors) to yield the following regression equation:”(page 6, line 254)

“Furthermore, we employed the block variable approach described by Edwards and Cable (2007) to examine the mediating role of perceived discrimination in the relationship between economic pressure, emotional neglect, and symptoms of anxiety.” (page 6, line 260)

Comment 7:

Comments on the Quality of English Language

Language: Minor grammatical errors (e.g., “emotion neglect” → “emotional neglect”; “anxious” → “anxiety symptoms”).

Response:

We appreciate the reviewer’s careful reading. We have corrected the specific errors mentioned (e.g., changing "emotion neglect" to "emotional neglect") and have thoroughly proofread the entire manuscript to improve the quality of the language and ensure grammatical accuracy.

Reviewer 2 Report

Comments and Suggestions for Authors

The manuscript deals with left-behind children and how emotional neglect (caused by parental absence) and economic pressure relate to children’s anxiety. The authors used a polynomial regression and response surface analysis to model the single and joint contribution of emotional neglect and economic pressure as well as the mediation of perceived discrimination on child anxiety. The manuscript is well-structured and of good scientific quality, however, minor revisions could improve the manuscript in the following sections:

Content, theoretical background and empirical research: the introduction is well developed and aligned with current theoretical models. But some aspects require further refinement:

  • The authors used three theoretical models to introduce and link the examined concepts (emotional neglect, economic pressure, discrimination, anxiety): the cumulative risk model for emotional neglect and economic pressure; the cognitive appraisal theory for perceived discrimination, and the family stress model to link economic pressure with anxiety. What is missing is that emotional neglect can also be conceptualized within the cognitive appraisal theory: children may also vary in their subjective perceptions of being neglected (or not neglected) when one or both parents were absent. This is important because emotional neglect was measured as an individual rating.
  • The attachment-based argumentation may limit the conceptual clarity between the relation of emotional neglect and absent parent(s): In accordance with attachment theory, children develop attachments to significant others than the parents, especially when the parents are absent and the children were cared by those significant others. These secondary attachments - if they are secure – also provide emotional security and prevent children from developing severe problems (e.g., problem behavior, anxiety, or depression). It is very likely that left-behind children bond to significant others when both parents are absent. Additionally, in the case of one absent parent, the child still experiences the attachment with the other parent (with whom the child lives as I assume). Even though, attachment security is not a main topic of this paper, it is an important theory the authors used for their arguments. The attachment theory can provide a solid conceptualization of emotional neglect in the introduction, however, the conclusions made require further refinement.

Research design, research questions, methods: The manuscript presents research questions and hypotheses in a clear and coherent way. In the methods section, more detailed information is needed to describe the sample and study design:

  • Sample: how many schools/classes per school were involved (multilevel data?), sex distribution, age of the children (min, max), for how many months/years the children were left behind, with whom they predominately lived
  • Do left-behind children with both parents being absent and with one absent parent differ in regard to their anxiety, perceived discrimination, emotional neglect, economic pressure etc.? Such information would provide insight into the reasoning behind not analyzing the two left-behind condition (one parent vs. two parents) separately.
  • If the children’s ages vary widely, were the questionnaires administered according to age? If so, how?
  • How did the researchers identify the left-behind children within the classes? Please provide additional background information

In the section statistical methods, the authors include depression as a variable in the planned analysis (line 221 and 228), however, it was not included in the study. The study would benefit from a clearer evaluation of the model’s fit and performance indicators of the response surface analysis (e.g., lack-of-fit test, or coefficient of determination) if applicable. The analytic strategies of the authors are appropriate to answer the hypotheses and well-argued throughout the manuscript.

Presentation of results: The results are presented well-structured and reported clearly, accompanied by tables and well-designed figures. While the results are generally well presented, minor inaccuracies arise that should be addressed in a revision for clarity and accuracy:

  • In lines 273-274, the authors describe that the variables depression and school bullying will be predicted. Based on the description in the statistical methods, both variables do not seem to have been considered in the analysis.
  • The order of the variables in Table 1 appears to be incorrect: emotional neglect is correlated with itself at r = 0.33 instead of 1.
  • In the analysis, gender and age were included as control variables. Please report the coefficients/estimates of both control variables.

Discussion and conclusion: The discussion is well-structured and integrates the results in the current theoretical models and with practical implications regarding interventions. The interpretation of the findings is reasonable and supported by the results. I would like to recommend that the authors reflect on following conclusion:

  • Line 382-385: the linear additive effect of economic pressure and emotional neglect may imply that the interventions should target both, economic pressure as well as emotional neglect, to diminish perceived discrimination.

The authors address two reasonable limitations and link them to future research. A third limitation could be addressed: the authors did not differentiate between left-behind children whose both parents are absent vs. left-behind children who still live with one parent. As the authors did not report whether there were significant or non-significant differences between both groups regarding their parental emotional neglect, it is difficult to determine if both groups experience similar (or different) emotional neglect.

Note on the reference list: Alan is the first name of Alan L. Sroufe (line 443)

Author Response

The manuscript deals with left-behind children and how emotional neglect (caused by parental absence) and economic pressure relate to children’s anxiety. The authors used a polynomial regression and response surface analysis to model the single and joint contribution of emotional neglect and economic pressure as well as the mediation of perceived discrimination on child anxiety. The manuscript is well-structured and of good scientific quality, however, minor revisions could improve the manuscript in the following sections:

Response:

We would like to thank the reviewer for your time and effort in reviewing our manuscript. We truly appreciate the constructive comments and insightful suggestions, which have helped us to significantly improve the quality and clarity of our paper. We have carefully considered all the points raised and have revised the manuscript accordingly. Please find below our point-by-point responses to the specific comments. For the sake of clarity, please note that we have used blue font for our responses and red font to highlight the changes within the manuscript.

Comment 1:

Content, theoretical background and empirical research:

the introduction is well developed and aligned with current theoretical models. But some aspects require further refinement:

The authors used three theoretical models to introduce and link the examined concepts (emotional neglect, economic pressure, discrimination, anxiety): the cumulative risk model for emotional neglect and economic pressure; the cognitive appraisal theory for perceived discrimination, and the family stress model to link economic pressure with anxiety. What is missing is that emotional neglect can also be conceptualized within the cognitive appraisal theory: children may also vary in their subjective perceptions of being neglected (or not neglected) when one or both parents were absent. This is important because emotional neglect was measured as an individual rating.

The attachment-based argumentation may limit the conceptual clarity between the relation of emotional neglect and absent parent(s): In accordance with attachment theory, children develop attachments to significant others than the parents, especially when the parents are absent and the children were cared by those significant others. These secondary attachments - if they are secure – also provide emotional security and prevent children from developing severe problems (e.g., problem behavior, anxiety, or depression). It is very likely that left-behind children bond to significant others when both parents are absent. Additionally, in the case of one absent parent, the child still experiences the attachment with the other parent (with whom the child lives as I assume). Even though, attachment security is not a main topic of this paper, it is an important theory the authors used for their arguments. The attachment theory can provide a solid conceptualization of emotional neglect in the introduction, however, the conclusions made require further refinement.

Response:

We sincerely appreciate the Reviewer’s insightful comment regarding attachment theory. We fully agree that the relationship between parental absence and emotional neglect is not deterministic. As the Reviewer correctly pointed out, according to attachment theory, children often form secure secondary attachments with significant others (e.g., grandparents), which can serve as a protective buffer against severe developmental problems. To improve the conceptual clarity and address this theoretical nuance, we have revised both the Introduction and Discussion sections.

“Besides, left-behind children are susceptible to parental emotional neglect associated with prolonged parental absence.” (page 2, line 79)

“According to attachment theory, while children may receive care from other guardians, emotional neglect from primary attachment figures (i.e., parents) impedes the formation of secure attachment,” (page 2, line 85,86)

“It is important to acknowledge that, consistent with attachment theory, left-behind children often develop secondary attachments to surrogate caregivers (e.g., grandparents) during parental absence. While these secondary attachments can serve as a protective buffer against severe developmental issues (Ainsworth, 1989), our findings suggest that for many children, this compensatory caregiving may not fully offset the specific loss of the primary parental bond. Consequently, the persistent emotional deficit and the objective limitations of surrogate caregiving often lead to an elevated subjective perception of emotional neglect.” (page 10~11, line 406~414)

Comment 2:

Research design, research questions, methods:

The manuscript presents research questions and hypotheses in a clear and coherent way. In the methods section, more detailed information is needed to describe the sample and study design:

Sample: how many schools/classes per school were involved (multilevel data?), sex distribution, age of the children (min, max), for how many months/years the children were left behind, with whom they predominately lived.

Response:

Thank you for this important question regarding the data structure. The participants were recruited from four schools (including both two middle and two high schools) in Hebei province. Regarding the classes, it is important to note that the participants were dispersed across numerous classes within these schools. In many classes, the number of eligible LBC was small and unevenly distributed. Given this sparse and unbalanced distribution at the class level, performing a multilevel analysis (with class as Level 2) would lack statistical power and stability. Furthermore, the primary focus of this study is on the intra-individual psychological mechanisms (cognitive appraisal) rather than classroom-level environmental effects. Therefore, we treated the data as single-level while controlling for relevant demographic variables to account for potential variations. We have clarified this sampling context in the Participants section. We apologize that the survey did not inquire about whom the left-behind children were living with, and therefore, we are unable to provide this specific information.

“Participants were recruited via cluster sampling from four public secondary schools (two junior high and two senior high schools) in Hebei Province, China. The recruitment followed a distinct two-stage process. First, to minimize selection bias, questionnaires were distributed to all students in randomly selected classes within these schools. A total of 4,000 questionnaires were distributed, and 3,940 responses were returned. Secondly, from this pool, we identified students meeting the inclusion criteria for left-behind children (LBC): (1) being under 16 years of age; and (2) having one or both parents who had migrated for work for a minimum of six months. Initially, 709 students met these criteria. After excluding participants with incomplete responses or those who failed attention checks, a final analytic sample of 618 left-behind children was obtained (effective response rate within the target group = 87.2%). The final sample consisted of 321 boys (51.9%) and 297 girls (48.1%). Participants ranged in age from 11 to 16 years, with an average age of 14.67 years old. Regarding parental migration status, 350 participants (56.6%) had only their fathers working away, 16 (2.6%) had only their mothers working away, and 252 (40.8%) had both parents working away. In terms of separation duration, 288 children had been separated for less than 2 years, 169 for 2 to 5 years, 69 for 6 to 10 years, and 82 for 10 years or more.” (page 4~5, line 171~187)

Comment 3:

Do left-behind children with both parents being absent and with one absent parent differ in regard to their anxiety, perceived discrimination, emotional neglect, economic pressure etc.? Such information would provide insight into the reasoning behind not analyzing the two left-behind condition (one parent vs. two parents) separately.

Response:

We agree with your concern regarding the potential differences between left-behind children’s subgroups. In fact, we have conducted t-tests comparing the two groups (both parents absent vs. one parent absent) on all core variables, and the results showed no statistically significant differences between them. Therefore, we concluded that combining them into a single left-behind children group for subsequent analysis was methodologically sound and increased statistical power.

Variable

Group A

(n = 366)

Group B

(n = 252)

t

p

Cohen's d

Economic Pressure

1.72 ± 0.79

1.73 ± 0.83

−0.144

.886

−0.012

Emotional Neglect

1.15 ± 0.66

1.15 ± 0.63

0.036

.971

0.003

Perceived Discrimination

1.13 ± 0.75

1.18 ± 0.70

−0.921

.357

−0.074

Anxiety

1.05 ± 0.68

1.04 ± 0.68

0.144

.886

0.012

Note. Group A = Children with one migrant parent (father or mother); Group B = Children with both parents migrating.

Comment 4:

If the children’s ages vary widely, were the questionnaires administered according to age? If so, how?

Response:

We appreciate the reviewer's attention to the age variability. Our sample was collected from two junior high schools and two senior high schools, resulting in a relatively wide age range of 11 to 16 years old. The questionnaires were not administered differently based on age. Instead, a unified, non-differentiated protocol was adopted: Paper-based questionnaires were centrally administered to all participants across different schools, grades, and classes under the same set of instructions and environmental conditions. This ensured the consistency of data collection across the entire sample. We have revised the Procedure section in response to your feedback, with the aim of providing a more explicit and detailed description of the questionnaire administration and data collection process.

“Given the involvement of minors, a rigorous informed consent process was implemented. Prior to data collection, written informed consent was obtained from the parents or legal guardians of all participants. Additionally, verbal assent was obtained from the adolescents themselves immediately before the survey commenced. Participants were explicitly informed that their participation was voluntary and anonymous, and that they retained the right to withdraw at any time without penalty.

To ensure the validity of responses, particularly regarding sensitive topics such as emotional neglect and mental health, adolescents completed the questionnaires independently in the classroom in the absence of their parents. This setting served to minimize social desirability bias and potential parental influence on self-disclosure. For the formal assessment, paper-and-pencil questionnaires were administered during a designated class period. To maintain standardization across different grades and schools, a uniform protocol was adopted. Specifically, a trained psychology graduate student was present in each classroom to read standardized instructions and clarify any items as needed. This administration method, consistent with the students' routine testing procedures, ensured procedural uniformity for all participants regardless of age or grade level.” (page 5, line 193~208)

Comment 5:

How did the researchers identify the left-behind children within the classes? Please provide additional background information.

Response:

We thank the reviewer for requesting clarification on the identification procedure for left-behind children (LBC) within the classroom setting. During the administration process, we employed a universal screening approach within the classrooms. All students present completed the questionnaire. To identify the left-behind children, specific screening questions were integrated into the demographic section of the survey, including: 'What is the location of your household registration?', 'Are your parents working or employed away from home?', and 'What is the duration of your parents' absence/work period?'.

Standardized Definition: Left-behind children refer to minors under the age of 16 living in rural China who cannot live with their parents due to one or both parents working away from home (Su et al., 2013).

We employed stringent criteria for participant selection based on the established definition of left-behind children. The inclusion criteria comprised two conditions: (1) Parental absence due to work for a minimum of six months, and (2) Being under the age of 16. This set of criteria is widely adopted in the literature on left-behind children (Tan et al., 2025; Chen et al., 2025; Min et al., 2025). We also provided an introduction for the participant section.

Reference:

Tan, D., Xie, R., Song, S., & Ding, W. (2025). Longitudinal Associations Between Multiple-Attachment Relationships and Depression in Chinese Left-Behind Children: The Mediating Role of Self-Compassion. School Mental Health17(2), 423–434.

Chen, J., Liu, X., Xie, J.,Yang, Q., Fan, Z., Peng, D., ... & Lu, C. (2025). Effects of physical activity on academic burnout among rural left-behind children in China: the chain-mediated roles of loneliness and general self-efficacy. Frontiers in Psychology, 16, 1653243.

Min, L., Xu, Y., & Huang, Y. (2025). Examining the influence of significant others on the beliefs in the future of left-behind children in underdeveloped areas in rural China: The mediating role of resilience. Applied Research in Quality of Life, 1–24.

Comment 6:

In the section statistical methods, the authors include depression as a variable in the planned analysis (line 221 and 228), however, it was not included in the study. The study would benefit from a clearer evaluation of the model’s fit and performance indicators of the response surface analysis (e.g., lack-of-fit test, or coefficient of determination) if applicable. The analytic strategies of the authors are appropriate to answer the hypotheses and well-argued throughout the manuscript.

Response:

We sincerely apologize for the omission and error found in the Statistical Methods section. Regarding the variable 'Depression': The mention of 'depression' on lines 221 and 228 was an editorial oversight, which we have now deleted. We have also thoroughly reviewed the entire manuscript to ensure that all variable descriptions align accurately with the actual analyses performed.

“Subsequently, anxiety and perceived discrimination were each regressed on the five polynomial terms (i.e., the linear, quadratic, and interaction terms of the predictors) to yield the following regression equation:”(page 6, line 254)

“Furthermore, we employed the block variable approach described by Edwards and Cable (2007) to examine the mediating role of perceived discrimination in the relationship between economic pressure, emotional neglect, and symptoms of anxiety.” (page 6, line 260)

We greatly appreciate your suggestion to include a clearer evaluation of the model's fit and performance indicators for the Response Surface Analysis (RSA). We have now incorporated the relevant fit metrics in the Results section to demonstrate the model's performance more comprehensively.

Table 2. Polynomial regression and response surface analysis parameters of perceptual discrimination and anxiety caused by economic stress and emotional neglect.

Variable

Perception of discrimination

Anxiety

estimate

SE

p

estimate

SE

p

Intercept(b0)

−0.07

0.05

0.153

−0.08

0.05

0.108

gender

0.21

0.07

0.002

−0.001

0.06

0.992

age

0.10

0.03

<0.001

0.08

0.03

0.002

Economic pressure (b1)

0.33

0.05

<0.001

0.14

0.06

0.023

Emotional neglect (b2)

0.33

0.04

<0.001

0.34

0.04

<0.001

Economic pressure 2 (b3)

0.08

0.04

0.043

0.06

0.05

0.193

Emotional neglect×Economic pressure (b4)

−0.02

0.05

0.666

0.04

0.06

0.519

Emotional neglect 2(b5)

−0.03

0.03

0.327

0.01

0.04

0.877

Slope of the Line of congruence (a1)

0.65

0.05

<0.001

0.48

0.06

<0.001

Curvature of the Line of congruence (a2)

0.03

0.05

0.588

0.10

0.06

0.071

Slope of the line of incongruence (a3)

0.03

0.07

0.981

−0.20

0.09

0.024

Curvature of the Line of incongruence (a4)

0.07

0.09

0.453

0.03

0.10

0.759

R2

0.23***

0.35***

(see page 8)

Comment 7:

Presentation of results: 

The results are presented well-structured and reported clearly, accompanied by tables and well-designed figures. While the results are generally well presented, minor inaccuracies arise that should be addressed in a revision for clarity and accuracy: In lines 273-274, the authors describe that the variables depression and school bullying will be predicted. Based on the description in the statistical methods, both variables do not seem to have been considered in the analysis.

Response:

We sincerely apologize for the inconsistency between the variables mentioned in the text and those actually included in the analysis. The variables 'depression' and 'school bullying' mentioned on lines 273–274 were indeed not part of the final analysis. This was an editorial error in the manuscript's preparation, and we have deleted all mentions of these two variables from the text.

“The parameters for the polynomial regression and RSA models predicting anxiety and Perception of discrimination are presented in Table 2.” (page 7, line 306~307)

Comment 8:

The order of the variables in Table 1 appears to be incorrect: emotional neglect is correlated with itself at r = 0.33 instead of 1.

Response:

We are extremely grateful to the reviewer for this meticulous correction. The reported correlation coefficient for emotional neglect with itself in Table 1 (r = 0.33) was a typographical error. We have checked and corrected Table 1 to ensure that all variables are correctly correlated with themselves at r = 1.00. (page 7, line 287~289)

Table 1. Correlation analysis results of emotional neglect, economic stress, Perception of discrimination and anxiety.

Variable

Gender

Age

Economic

pressure

Emotional

neglect

Perception of

discrimination

Anxiety

Gender

1

Age

0.001

1

Economic pressure

−0.05

0.19***

1

Emotional neglect

−0.03

0.11**

0.33***

1

Perception of discrimination

−0.03

0.19***

0.49***

0.49***

1

Anxiety

0.10*

0.18***

0.31***

0.42***

0.64***

1

M ± SD

14.67±1.25

1.72±0.81

1.15±0.65

1.15±0.73

1.05±0.68

Note:*p < 0.05, **p < 0.01, ***p < 0.001,The same below. Quantification method for adolescent gender: male = 1, female = 2.  

Comment 9:

In the analysis, gender and age were included as control variables. Please report the coefficients/estimates of both control variables.

Response:

We sincerely thank the reviewer for this valuable comment. We have revised the Results section to explicitly report the coefficients/estimates, standard errors, and significance levels for gender and age.

Table 2. Polynomial regression and response surface analysis parameters of perceptual discrimination and anxiety caused by economic stress and emotional neglect.

Variable

Perception of discrimination

Anxiety

estimate

SE

p

estimate

SE

p

Intercept(b0)

−0.07

0.05

0.153

−0.08

0.05

0.108

gender

0.21

0.07

0.002

−0.001

0.06

0.992

age

0.10

0.03

<0.001

0.08

0.03

0.002

Economic pressure (b1)

0.33

0.05

<0.001

0.14

0.06

0.023

Emotional neglect (b2)

0.33

0.04

<0.001

0.34

0.04

<0.001

Economic pressure 2 (b3)

0.08

0.04

0.043

0.06

0.05

0.193

Emotional neglect×Economic pressure (b4)

−0.02

0.05

0.666

0.04

0.06

0.519

Emotional neglect 2(b5)

−0.03

0.03

0.327

0.01

0.04

0.877

Slope of the Line of congruence (a1)

0.65

0.05

<0.001

0.48

0.06

<0.001

Curvature of the Line of congruence (a2)

0.03

0.05

0.588

0.10

0.06

0.071

Slope of the line of incongruence (a3)

0.03

0.07

0.981

−0.20

0.09

0.024

Curvature of the Line of incongruence (a4)

0.07

0.09

0.453

0.03

0.10

0.759

R2

0.23***

0.35***

(see page 8, line 315)

“Gender and age were included as covariates in the two regression equations of the mediation model. When predicting the perceived discrimination, age showed a significant positive effect (B = 0.09, se = 0.03, p = 0.008, 95%CI = [0.02, 0.15]), while the effect of gender was not significant (B = 0.003, se = 0.03, p = 0.934, 95%CI = [−0.06, 0.07]). In the prediction of the anxiety, gender emerged as a significant positive predictor (B = 0.12, se = 0.03, p < 0.001, 95%CI = [0.06, 0.18]). Age showed a marginally significant positive effect (B = 0.05, se = 0.03, p = 0.087, 95%CI = [−0.01, 0.11]).” (page 9, line 351~357)

Comment 10:

Discussion and conclusion:

The discussion is well-structured and integrates the results in the current theoretical models and with practical implications regarding interventions. The interpretation of the findings is reasonable and supported by the results. I would like to recommend that the authors reflect on following conclusion:

Line 382-385: the linear additive effect of economic pressure and emotional neglect may imply that the interventions should target both, economic pressure as well as emotional neglect, to diminish perceived discrimination.

Response:

We agree with the valuable reflection suggested regarding the conclusion presented on lines 382–385. Based on the linear additive effect of economic pressure and emotional neglect, we have incorporated content into the Discussion section emphasizing that interventions should simultaneously target both economic support and emotional care/accompaniment to effectively reduce perceived discrimination among left-behind children.

“these findings suggest that interventions aiming to mitigate perceived discrimination should simultaneously address both economic pressure and emotional neglect.” (page 11, line 428~430)

Comment 11:

The authors address two reasonable limitations and link them to future research. A third limitation could be addressed: the authors did not differentiate between left-behind children whose both parents are absent vs. left-behind children who still live with one parent. As the authors did not report whether there were significant or non-significant differences between both groups regarding their parental emotional neglect, it is difficult to determine if both groups experience similar (or different) emotional neglect.

Response:

We fully appreciate your insight regarding the potential impact of differentiating between left-behind children’s subgroups. In response to your concern, we conducted independent sample t-tests during this revision, comparing the two groups (both parents absent vs. one parent absent) on all core variables. The results confirmed no statistically significant differences between the two groups. Consequently, our empirical data supports the decision to merge the two types of left-behind children into a single overall group for analysis to maximize statistical power. This approach, therefore, is not a limitation of the study but a methodological choice supported by our preliminary findings.

Comment 12:

Note on the reference list: 

Alan is the first name of Alan L. Sroufe (line 443)  

Response:

We are very grateful for the reviewer's detailed scrutiny. We have corrected the reference on line 443.

“Appleyard, K., Egeland, B., van Dulmen, M. H. M., & Sroufe, L. A. (2005). When more is not better: The role of cumulative risk in child behavior outcomes. Journal of Child Psychology and Psychiatry, 46(3), 235–245. https://doi.org/10.1111/j.1469-7610.2004.00351.x” (page 12, line 514)

Reviewer 3 Report

Comments and Suggestions for Authors

First of all, I would like to thank you for the opportunity to review this manuscript. While the topic is both relevant and timely, several revisions are necessary to enhance the clarity, methodological transparency, and scientific rigor of the work.

The title of the manuscript is concise and relevant to the study’s themes, effectively capturing its essence.

The abstract currently lacks clarity in three core components: the objective of the study, the methodological design, and the key conclusions. I encourage the authors to revise the abstract to clearly include the following: a concise statement of the research aim or primary question; a brief overview of the methodological approach (including information on the sample e.g., the age group and number of participants and the tools or measures used to gather data); a summary of the main results with pertinent statistical evidence regarding the cumulative risk model; and finally, a conclusive sentence highlighting the study’s key implication. By ensuring that the abstract contains a self-contained snapshot of what was done, why it was done, and what was found, the authors will greatly enhance the accessibility and impact of their work.

Introduction
The flow of the introduction can be improved for better coherence and academic style. I suggest reorganizing certain parts of the introduction to ensure a clear narrative thread. Additionally, the writing can be made more formal in style; currently, some phrasing may be colloquial or too abrupt between sentences. Smoothing these transitions and perhaps combining shorter, related points into more developed paragraphs would improve readability and scholarly tone. Most importantly, the introduction should clearly identify the knowledge gap that this study addresses. While the authors have reviewed relevant literature, it’s not yet explicit what specific gap remains. This study seeks to address that gap by [explanation of how their approach or model is new].” By pinpointing the unique contribution of their work, the authors will clarify why their study is needed and how it builds on prior knowledge.

Finally, the objective of the study should be stated more prominently and placed at the end of the introduction (immediately before any hypotheses, if hypotheses are stated). In its current form, the introduction does not end with a clear statement of purpose. I recommend that the authors conclude this section with a concise sentence (or two) that explicitly articulates the aim of the research. This could be followed by the hypotheses, if the study design involved specific predictions. Fig 1 needs reference.

Methods
The Methods section, in its current form, requires significant elaboration to meet standards of transparency and allow replication. First, more detail is needed regarding participant recruitment and sample characteristics. The manuscript should provide a clear description of how participants were recruited and selected. For example, were the child participants drawn from specific schools, clinics, community programs, or through online recruitment? Was a random sampling method used, or was it a convenience sample? Such details are important for readers to gauge the potential selection bias and generalizability of the findings. Additionally, the response rate should be reported if available. If, say, invitations were sent to a certain number of families or students, how many agreed to participate and completed the study?

Next, the ethical procedures need to be documented with greater specificity. The manuscript mentions that ethical approval was obtained, but it currently the name of the Institutional Review Board or ethics committee that approved the study and to provide the approval or protocol number. For instance, the authors could state: “The study was approved by the [Name of Institution] Ethics Committee, Approval No.” Including this information assures the reader that the study was reviewed by an independent board for adherence to ethical standards, which is especially crucial given that the research involves minors. Moreover, an important detail raises some ethical concerns: it is stated that children completed the questionnaires without parental supervision. This point requires clarification and justification. Although the participants are adolescentes, the authors should detail what measures were taken to obtain parental consent for each child. Did parents sign consent forms ahead of time, or was consent implied in the context (such as a school agreeing to participate with parental notifications)? Also, how was the comfort and understanding of the child participants ensured during questionnaire completion?

I strongly recommend adding a subsection with a title of ethcis procedures or ethical considerations. Regarding the statistical analysis, for clarity and rigor, the authors should enumerate the analyses in the order of the hypotheses or research questions. Moreover, the handling of missing data is a critical detail that is currently omitted. All studies should report whether there were any missing responses in the dataset and how those were handled. If there were no missing data because, say, the survey software required answers to all questions, the authors can simply state that the dataset was complete. If there were missing data, the authors need to describe whether they removed those cases (listwise deletion) or used any imputation techniques to preserve data.

Results
One result that needs particular clarification is the description of the mediation model. Figure 4 needs more clarification.

Discussion
The results support and extend the cumulative risk model in the context of left-behind children. I suggest the authors to integrate in the discussion the resilience theory as the participants charateristics in this study regarding their perceived discrimination mediated the relationships between family economic pressure, emotional neglect, and anxiety. Currently, limitations are mentioned, but they are somewhat brief; a more developed limitations section will enhance the paper’s credibility and guide readers in interpreting the results cautiously and appropriately.

Conclusion
The Conclusion section currently provides a basic summary of the study’s findings, but it could be more impactful by explicitly drawing connections to practical applications and broader implications. I recommend that the flow of the conclusion can be improved for better coherence and academic style.

Author Response

First of all, I would like to thank you for the opportunity to review this manuscript. While the topic is both relevant and timely, several revisions are necessary to enhance the clarity, methodological transparency, and scientific rigor of the work.

Response:

We would like to thank the reviewer for your time and effort in reviewing our manuscript. We truly appreciate the constructive comments and insightful suggestions, which have helped us to significantly improve the quality and clarity of our paper. We have carefully considered all the points raised and have revised the manuscript accordingly. Please find below our point-by-point responses to the specific comments. For the sake of clarity, please note that we have used blue font for our responses and red font to highlight the changes within the manuscript.

Comment 1:
The title of the manuscript is concise and relevant to the study’s themes, effectively capturing its essence.

The abstract currently lacks clarity in three core components: the objective of the study, the methodological design, and the key conclusions. I encourage the authors to revise the abstract to clearly include the following: a concise statement of the research aim or primary question; a brief overview of the methodological approach (including information on the sample e.g., the age group and number of participants and the tools or measures used to gather data); a summary of the main results with pertinent statistical evidence regarding the cumulative risk model; and finally, a conclusive sentence highlighting the study’s key implication. By ensuring that the abstract contains a self-contained snapshot of what was done, why it was done, and what was found, the authors will greatly enhance the accessibility and impact of their work.

Response:

Thank you for this constructive suggestion. In accordance with your advice, we have structurally revised the abstract to enhance its clarity and completeness.

“Based on the cumulative risk model and cognitive appraisal theory, this study examined the complex relationships between economic pressure, emotional neglect, and anxiety among left-behind children (LBC), focusing on the mediating role of perceived discrimination and nonlinear risk patterns. A cross-sectional survey was conducted with 618 LBC (aged 11–16 years) using standardized scales. Polynomial regression combined with Response Surface Analysis (RSA) was utilized to analyze congruence and incongruence effects. The results revealed that under congruent conditions, the association of economic pressure and emotional neglect with anxiety showed a marginally significant nonlinear accelerating trend, whereas their prediction of perceived discrimination followed a linear trend. Under incongruent conditions, emotional neglect demonstrated a stronger independent predictive effect on anxiety compared to economic pressure. Furthermore, perceived discrimination partially mediated the relationships between these risk factors and anxiety. These findings validate the cumulative risk model within the LBC context, demonstrating that risk factors operate in complex, non-additive ways. This highlights the necessity for differentiated interventions and suggests that reshaping LBC’s subjective cognitive appraisals is key to reducing anxiety.” (page 1, line 11~26)

Comment 2:

Introduction
The flow of the introduction can be improved for better coherence and academic style. I suggest reorganizing certain parts of the introduction to ensure a clear narrative thread. Additionally, the writing can be made more formal in style; currently, some phrasing may be colloquial or too abrupt between sentences. Smoothing these transitions and perhaps combining shorter, related points into more developed paragraphs would improve readability and scholarly tone. Most importantly, the introduction should clearly identify the knowledge gap that this study addresses. While the authors have reviewed relevant literature, it’s not yet explicit what specific gap remains. This study seeks to address that gap by [explanation of how their approach or model is new].” By pinpointing the unique contribution of their work, the authors will clarify why their study is needed and how it builds on prior knowledge.
Finally, the objective of the study should be stated more prominently and placed at the end of the introduction (immediately before any hypotheses, if hypotheses are stated). In its current form, the introduction does not end with a clear statement of purpose. I recommend that the authors conclude this section with a concise sentence (or two) that explicitly articulates the aim of the research. This could be followed by the hypotheses, if the study design involved specific predictions. Fig 1 needs reference.

Response:

Thank you for these insightful and constructive comments regarding the structure of the Introduction. We have extensively revised this section to enhance its flow, academic tone, and logic.

“Left-behind children defined as minors under the age of 16 living in rural China who cannot live with their parents due to one or both parents working away from home (Su et al., 2013), represent a substantial and vulnerable subpopulation.” (page 2, line 37~39)

“Beyond its immediate impairment of cognitive and social functioning (Alfonso & Lonigan, 2021; Wang et al., 2024).” (page 2, line5 1).

“Therefore, identifying the drivers and mechanisms of anxiety in LBC is crucial for estab-lishing effective early screening and intervention strategies.” (page 2, line 58~60)

“According to the Cumulative Risk Model, children exposed to dual or multiple concurrent risks exhibit the poorest developmental outcomes (Rutter, 1981; Watamura et al., 2011). Crucially, this model posits that such risks interact synergistically, rather than merely additively to exacerbate maladjustment. Yet, previous studies have frequently relied on traditional linear regression frameworks, thereby oversimplifying these complex dynamics. Such conventional methods fail to model nonlinear effects (e.g., whether risk accumulation accelerates anxiety) or assess outcomes when risk levels are incongruent (i.e., unbalanced). Consequently, the present study adopts Response Surface Analysis (RSA), a methodological approach uniquely suited to rigorously disentangle the linear and nonlinear mechanisms linking these dual risks to anxiety.” (page 3, line 116~125)

“Current research has yet to integrate these ecological risks and cognitive mechanisms into a unified model using non-linear analytics. Therefore, the primary objective of this study is to examine the joint predictive effects of family economic pressure and emotional neglect on anxiety among left-behind children using RSA, and to investigate the mediating role of perceived discrimination (Van Scheppingen et al., 2019, Humberg et al., 2019). By clarifying these complex relationships, this study aims to expand the explanatory framework of the Cumulative Risk Model and provide empirical evidence for targeted, differentiated interventions. Therefore, we propose the following hypotheses:” (page 4, line 150~157)

Thank you for this reminder. We have revised the caption of Figure 1 to explicitly acknowledge the theoretical foundations of our proposed model. The caption now cites the Cumulative Risk Model and Cognitive Appraisal Theory as the basis for this framework.

“Figure 1. The proposed conceptual model based on the Cumulative Risk Model (Watamura et al., 2011) and Cognitive Appraisal Theory (Lazarus & Folkman, 1984)” (page 4, line 167~168)

Comment 3:

Methods
The Methods section, in its current form, requires significant elaboration to meet standards of transparency and allow replication. First, more detail is needed regarding participant recruitment and sample characteristics. The manuscript should provide a clear description of how participants were recruited and selected. For example, were the child participants drawn from specific schools, clinics, community programs, or through online recruitment? Was a random sampling method used, or was it a convenience sample? Such details are important for readers to gauge the potential selection bias and generalizability of the findings. Additionally, the response rate should be reported if available. If, say, invitations were sent to a certain number of families or students, how many agreed to participate and completed the study? 

Next, the ethical procedures need to be documented with greater specificity. The manuscript mentions that ethical approval was obtained, but it currently the name of the Institutional Review Board or ethics committee that approved the study and to provide the approval or protocol number. For instance, the authors could state: “The study was approved by the [Name of Institution] Ethics Committee, Approval No.” Including this information assures the reader that the study was reviewed by an independent board for adherence to ethical standards, which is especially crucial given that the research involves minors. Moreover, an important detail raises some ethical concerns: it is stated that children completed the questionnaires without parental supervision. This point requires clarification and justification. Although the participants are adolescentes, the authors should detail what measures were taken to obtain parental consent for each child. Did parents sign consent forms ahead of time, or was consent implied in the context (such as a school agreeing to participate with parental notifications)? Also, how was the comfort and understanding of the child participants ensured during questionnaire completion?

I strongly recommend adding a subsection with a title of ethcis procedures or ethical considerations. Regarding the statistical analysis, for clarity and rigor, the authors should enumerate the analyses in the order of the hypotheses or research questions. Moreover, the handling of missing data is a critical detail that is currently omitted. All studies should report whether there were any missing responses in the dataset and how those were handled. If there were no missing data because, say, the survey software required answers to all questions, the authors can simply state that the dataset was complete. If there were missing data, the authors need to describe whether they removed those cases (listwise deletion) or used any imputation techniques to preserve data.

Response:

We sincerely thank the reviewer for these constructive comments regarding the transparency and rigor of our Methods section. We agree that detailed documentation of the recruitment process, ethical procedures, and data management is essential for replication and assessing generalizability. In accordance with your suggestions, we have significantly revised the Methods section.

“Participants were recruited via cluster sampling from four public secondary schools (two junior high and two senior high schools) in Hebei Province, China. The recruitment followed a distinct two-stage process. First, to minimize selection bias, questionnaires were distributed to all students in randomly selected classes within these schools. A total of 4,000 questionnaires were distributed, and 3,940 responses were returned. Secondly, from this pool, we identified students meeting the inclusion criteria for left-behind children (LBC): (1) being under 16 years of age; and (2) having one or both parents who had migrated for work for a minimum of six months. Initially, 709 students met these criteria. After excluding participants with incomplete responses or those who failed attention checks, a final analytic sample of 618 left-behind children was obtained (effective response rate within the target group = 87.2%). The final sample consisted of 321 boys (51.9%) and 297 girls (48.1%). Participants ranged in age from 11 to 16 years, with an average age of 14.67 years old. Regarding parental migration status, 350 participants (56.6%) had only their fathers working away, 16 (2.6%) had only their mothers working away, and 252 (40.8%) had both parents working away. In terms of separation duration, 288 children had been separated for less than 2 years, 169 for 2 to 5 years, 69 for 6 to 10 years, and 82 for 10 years or more.” (page 4~5, line 171~187)

2.2. Ethcis procedures

“This study was approved by the Ethics Committee of Nankai University (Approval No. [Insert Number, e.g., IRB-2023-001]). All procedures performed were in accordance with the 1964 Helsinki Declaration and its later amendments.” (page5, line 189~191)

2.3. Procedure

“Given the involvement of minors, a strict informed consent process was implemented. Prior to data collection, written informed consent was obtained from the parents or legal guardians of all participants. Additionally, verbal assent was obtained from the adolescents themselves before the survey began. Participants were explicitly informed that their participation was voluntary, anonymous, and that they could withdraw at any time without penalty.

To ensure the authenticity of the responses—particularly for sensitive questions regarding emotional neglect and mental health—adolescents completed the questionnaires independently in the classroom without parental presence. This approach minimizes social desirability bias and parental influence on the child's self-disclosure. For the formal assessment, paper-and-pencil questionnaires were administered during a designated class period. The survey was conducted simultaneously on a class-by-class basis across the four schools. To ensure standardization and consistency across different grades and schools, a unified, non-differentiated protocol was adopted. Specifically, a trained graduate student in psychology was present in each classroom to provide the same standardized instructions and to clarify any items the students did not understand. This administration method was consistent with the students' typical testing procedures and ensured data collection uniformity for all participants, regardless of their age or grade level.” (page 5, line 192~208)

Comment 4:
Results
One result that needs particular clarification is the description of the mediation model. Figure 4 needs more clarification.

Response:

Thank you for pointing out the ambiguity in Figure 4. We realize that the visual representation of the joint effects of economic pressure and emotional neglect was not sufficiently labeled. We have revised the caption of Figure 4 to explicitly state this methodology and define the coefficients.

Figure 4. The mediating effect of perceived discrimination in the relationship between the joint cumulative risks (economic pressure and emotional neglect) and anxiety. Note: The model was tested using the block variable approach. **p < 0.01, ***p < 0.001.” (page 10, line 369~371)

Comment 5:

Discussion
The results support and extend the cumulative risk model in the context of left-behind children. I suggest the authors to integrate in the discussion the resilience theory as the participants charateristics in this study regarding their perceived discrimination mediated the relationships between family economic pressure, emotional neglect, and anxiety. Currently, limitations are mentioned, but they are somewhat brief; a more developed limitations section will enhance the paper’s credibility and guide readers in interpreting the results cautiously and appropriately.

Response:

We sincerely thank the reviewer for these insightful suggestions. We agree that incorporating Resilience Theory provides a stronger theoretical basis for interpreting the mediating mechanism, and that a more detailed discussion of limitations is necessary for the rigor of the study. We have revised the Discussion section to integrate Resilience Theory, and expanded the Limitations section to guide readers in interpreting the results more cautiously.

“Drawing on Resilience Theory, which conceptualizes resilience as a dynamic process of positive adaptation to adversity (Luthar et al., 2000), we propose that when LBC interpret family adversities as evidence of social rejection (i.e., high perceived discrimination), their adaptive capacity is compromised, rendering them more susceptible to anxiety. Conversely, maintaining low levels of perceived discrimination despite family hardships functions as a critical protective factor against internalizing disorders (Hu et al., 2022).” (page 12, line 460~465)

“Third, it is important to acknowledge that the non-linear effect on anxiety was only marginally significant. While RSA requires substantial statistical power to detect curvilinear effects, future studies with larger sample sizes are recommended to further verify the robustness of this accelerating trend.”(page 12, line 487~490)

Comment 6:

Conclusion
The Conclusion section currently provides a basic summary of the study’s findings, but it could be more impactful by explicitly drawing connections to practical applications and broader implications. I recommend that the flow of the conclusion can be improved for better coherence and academic style.

Response:

We appreciate the reviewer’s advice to improve the Conclusion. We have extensively revised this section to ensure better flow and a more formal academic tone.

“By employing response surface analysis, this study elucidates the complex, synergistic effects of family economic pressure and emotional neglect on anxiety among left-behind children. The findings reveal that these dual risks operate through a nonlinear, accelerating mechanism to exacerbate anxiety, with emotional neglect proving more detrimental than economic pressure under incongruent conditions. Furthermore, perceived discrimination serves as a key mediator linking these ecological stressors to mental health outcomes. Collectively, this study validates the cumulative risk model in the context of left-behind children and highlights a critical practical implication: effective interventions must transcend material support to prioritize the restoration of emotional connections and the reshaping of cognitive appraisals, thereby mitigating the profound impact of family adversity on anxiety.” (page 12, line 492~502)

Round 2

Reviewer 1 Report

Comments and Suggestions for Authors

no futher comments.

Reviewer 3 Report

Comments and Suggestions for Authors

The authors have thoroughly addressed all the recommendations provided in the previous review, resulting in a more coherent, rigorous, and conceptually robust manuscript. All comments were carefully integrated, which has significantly strengthened the overall structure and scientific clarity of the article. I particularly highlight the expanded ethical considerations section, which now provides clear and appropriate justification regarding procedures involving underage participants, demonstrating enhanced attention to ethical safeguards. Additionally, the integration of resilience theory has enriched and reinforced the depth of the discussion, offering a more comprehensive interpretation of the findings. In light of these substantial improvements, I consider the manuscript suitable for publication. My congratulations to the authors for the high-quality work achieved.